# DIVA: Discrete Diffusion Vision-Language-Action Models for Parallelized Action Generation

## Abstract

Vision-Language-Action (VLA) models have shown promising results in robot control, yet prevailing auto-regressive frameworks suffer from inherent limitations, such as error accumulation and temporal rigidity in action generation. To address this, we introduce a **DI**screte diffusion **V**ision-language-**A**ction model (DIVA), a discrete diffusion-based VLA framework that reformulates action generation as an iterative denoising process over discrete latent representations. The innovation of DIVA lies in the unified discrete diffusion architecture that systematically integrates three core designs: first, a learnable discrete action tokenization process bridges continuous action with the structural multimodal space. Second, A latent-driven policy learning strategy is proposed to align the representative space of the vision-language backbone and the policy head through a joint optimization. Third, a selective group unmasking strategy is introduced during the discrete diffusion decoding to preserve spatiotemporal coherence. Extensive evaluation demonstrates that DIVA achieves state-of-the-art performance in both simulated and real-world environments, validating its advantages in generating coherent, precise, and generalizable robot behaviors. Our work establishes a robust and scalable paradigm for future embodied decision-making systems.

## 1 Introduction

The rapid development of robotics has enabled embodied machines to perform increasingly diverse tasks, achieving notable success in structured environments such as industrial assembly lines, warehouse logistics, and controlled household chores. However, generalization to open-world, unstructured scenarios remains a fundamental challenge due to the complexity and variability of real-world settings. Concurrently, the emergence of large multimodal foundation models, such as CLIP Radford et al. (2021) for visual-language alignment, DINO Oquab et al. (2023) for semantic understanding, and R1 Guo et al. (2025) for reasoning ability incentivization, has demonstrated unprecedented perceptual and reasoning capabilities, creating new opportunities for general-purpose robotic systems. A growing trend of research Black et al. (2024); Kim et al. (2024); Bjorck et al. (2025) has sought to integrate the powerful capabilities of foundation models into robotic control frameworks, spurring the emergence of Vision-Language-Action (VLA) models as a prominent direction.

Vision-Language-Action (VLA) models bridge high-level instruction understanding with low-level motion generation, enabling robots to interpret and execute tasks from multimodal instructions in dynamic environments. (Zitkovich et al., 2023; Shridhar et al., 2022). By integrating perception, reasoning, and control within a unified architecture, VLAs aim to endow robotic systems with greater generalization and interactivity. Despite recent progress, it remains a challenging problem to translate perceptual and linguistic inputs into continuous and sequential controls for high quality action generation (Chi et al., 2023; Reed et al., 2022). Existing approaches largely fall into two categories (Ma et al., 2024; Zhong et al., 2025). The first category relies on the powerful multi-modal autoregressive transformers (Wu et al., 2023), which generate actions in a token-by-token sequential manner. While simple and widely adopted, this paradigm is inherently constrained by the sequential decoding strategy, which leads to issues such as error accumulation, temporal rigidity, and limited contextual flexibility (Ranzato et al., 2015). The second one employs continuous diffusion models, which excel at capturing multimodal action distributions through iterative refinement (Black et al.,

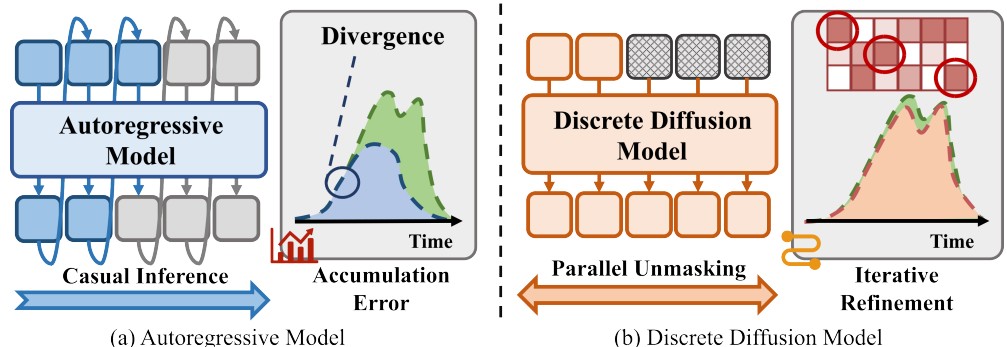

(a) Autoregressive Model      (b) Discrete Diffusion Model

Figure 1: Comparison between autoregressive and discrete diffusion models. (a) Autoregressive decoding suffers from error accumulation due to its rigid sequential nature. (b) Discrete diffusion enables parallel inference for iterative refinement, offering greater flexibility action decoding.

2023; Ren et al., 2024). However, these models often operate as decoupled decoders that are misaligned with the pretrained representations of the underlying vision-language backbone, leading to inefficiencies and training instability.

To address these limitations, we introduce DIVA, a unified VLA framework that reformulates action generation as a discrete diffusion process. Our approach is motivated by the need for a more expressive, robust, and natively-integrated decoding mechanism that remains consistent with large language model architectures. DIVA addresses three key challenges in current VLA methods. First, existing discretization approaches often rely on fixed binning strategies, which struggle to capture the nuanced structure of continuous action spaces (Brohan et al., 2022; Jang et al., 2022). To overcome this, we introduce a Discrete Action Tokenization module that learns an adaptive mapping from continuous action to a structured vocabulary, preserving fine-grained motion information. Second, the training of VLAs often suffers from a disconnect between the pretrained backbone and the policy head (Kim et al., 2024). We tackle this through Latent-Driven Policy Learning, which employs a latent regularization strategy to jointly optimize the discrete vision-language representations and continuous action. Third, conventional decoding processes in discrete diffusion models follow the confidence-first principle for selective token unmasking, which tends to generate actions in a fragmented, token-by-token manner and disrupt more complex temporal coherence (Austin et al., 2021a). To address this, we propose a Selective Group Unmasking strategy that enables group-level decoding to facilitate inter-action dependencies in the process of action generation. Together, these components enable DIVA to bridge the continuous action space with discrete reasoning, forming the unified vision-langauge-action framework for action generation in dynamic environments.

In summary, the contributions of this work are threefold:

- We propose the first discrete diffusion vision-language-action model that unifies discrete action tokenization and decoding.
- We introduce three core technical innovations to construct a unified framework: a discrete action tokenizer bridging the continuous action and language space, a latent-driven policy learning objective for joint optimization, and a selective group unmasking strategy ensuring temporally coherent decoding.
- Extensive simulated and real-world evaluations demonstrate state-of-the-art performance across multiple benchmarks, confirming the effectiveness of our method.

## 2 RELATED WORKS

### 2.1 VISION-LANGUAGE-ACTION MODELS

Research on Vision–Language–Action (VLA) models has converged along two complementary but distinct design axes: how actions are represented and generated and how tightly action generation

is integrated with large pretrained vision–language models (VLMs) (Ma et al., 2024; Zhong et al., 2025). One prevalent family frames control as a discrete sequence prediction problem, tokenizing low-level actions and decoding them autoregressively on top of a VLM (Brohan et al., 2022; Chen et al., 2021), which leverages strong sequence modeling and pretrained cross-modal features but suffers from serial decoding latency and compounding errors (Shridhar et al., 2023). Another family models entire action trajectories in continuous spaces and generates them via iterative refinement. Denoising diffusion and flow-matching methods (Chi et al., 2023; Hou et al., 2025) naturally capture multimodal and smooth behaviors but incur multiple inference steps and typically require separate modules and training schedules (Reed et al., 2022).

Recent research has focused on transferring the powerful perceptual and reasoning capabilities of large pretrained models to Vision–Language–Action (VLA) tasks, proposing a range of methods and frameworks to support this goal. Approaches include co-training (Black et al., 2024; Reed et al., 2022) as a training strategy and engineering techniques for runtime efficiency such as fast diffusion samplers (Chi et al., 2023; Hou et al., 2025). OpenVLA-OFT (Kim et al., 2025) improves inference efficiency and flexibility by combining parallel decoding, action chunking, and continuous action representations; Moto (Chen et al., 2024) leverages unsupervised latent motion tokens as an intermediate language to absorb motion priors from videos and transfer them to robot manipulation; VQ-VLA (Wang et al., 2025) scales action tokenization via a convolutional residual VQ-VAE with progressive training to achieve more stable representations; and DreamVLA (Zhang et al., 2025) incorporates world-knowledge forecasting into VLA models, predicting key environmental features rather than full future frames to enhance action planning.

## 2.2 Discrete Diffusion Large Language Models

Discrete Diffusion Large Language Models (dLLMs) have recently been proposed as an alternative to autoregressive generation by iteratively denoising masked sequences rather than producing tokens strictly left-to-right (Austin et al., 2021b; Lou et al., 2024). This paradigm supports bidirectional attention, parallel decoding, and iterative refinement, thereby improving reasoning, controllability, and infilling. Recent scaling efforts such as LLaDA (Nie et al., 2025) and LLaDA 1.5 (Zhu et al., 2025) demonstrate that dLLMs can match or surpass autoregressive baselines in text tasks, while multimodal extensions including LLaDA-V (You et al., 2025) and LaViDa (Li et al., 2025) highlight their applicability beyond language. Inspired by the powerful parallel decoding capabilities of previous work, DIVA introduces a unified discrete-diffusion framework that incorporates precise discrete tokenization and flexible decoding, yielding for efficient and scalable parallelized action generation for vision-language-action tasks.

## 3 Method

### 3.1 Preliminaries

Discrete diffusion large language models Nie et al. (2025); Zhu et al. (2025); You et al. (2025); Li et al. (2025); Yang et al. (2025)) introduce a novel generative approach that integrates discrete diffusion techniques Austin et al. (2021b) with conventional language modeling to establish a non-autoregressive generative paradigm. These methods construct a model distribution through a forward Markov process and its reverse process. Given an input sequence $x_0 = [x_0^1, x_0^2, ..., x_0^N] \sim p(x)$, the forward process gradually corrupts $x_0$ into an increasingly noisy state $x_t$, with each token independently being masked into a special token [MASK] in probability $t$, $t \in [0, 1]$. This process is formulated using Bernoulli variables $\{b_k\}_{k=1}^N \sim \text{Bernoulli}(t)$, as

$$x_t^k = \begin{cases} \texttt{[MASK]}, & b_k = 1, \\ x_0^k, & b_k = 0. \end{cases} \tag{1}$$

The reverse process employs a parametric mask predictor that recovers the marginal distribution $p_\theta$ from the noisy variables. Beginning with a fully-masked sequence $x_1$, the posterior $\hat{x}_0$ is progressively sampled through an iterative remasking strategy for refinement. For a given remasking ratio $\gamma \in (0, 1)$, the mask predictor concurrently assesses the likelihood of candidate predictions across all masked tokens. This process yields the intermediate state $\hat{x}_s$ ($s \in [0, t)$) through the remasking of a fraction $\gamma$ of the originally masked tokens. Then $x_s$ is refined in an iterative process until all

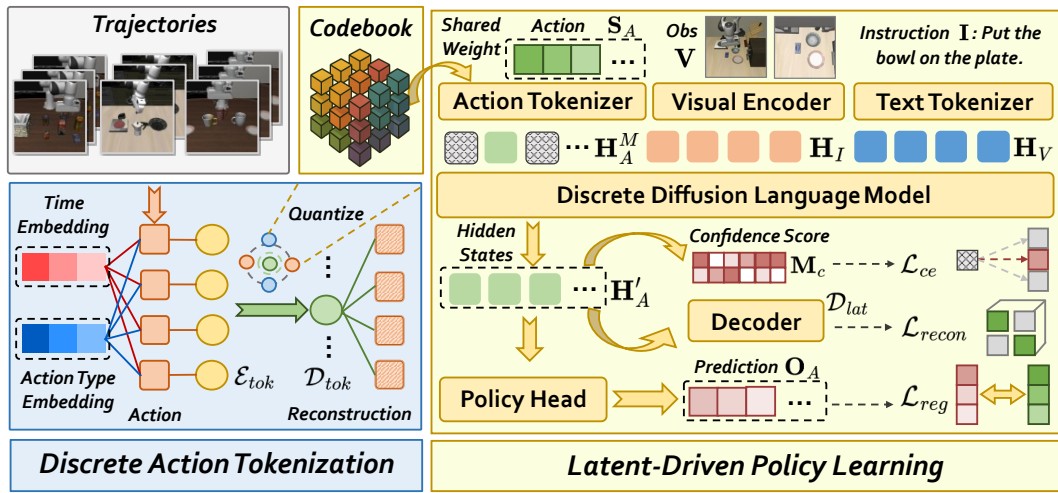

Figure 2: The architecture of DIVA. Our framework introduces two key innovations to enable precise and coherent action generation. First, a Discrete Action Tokenization process bridges continuous actions with structural language space. Second, a Latent-Driven Policy Learning paradigm jointly trains a discrete diffusion model with latent-space regularization, allowing iteratively refined parallel action decoding.

tokens are unmasked. Common unmasking strategies include low-confidence remasking and semi-autoregressive remasking. The overall transition from the fully-masked sequence $x_1$ to the posterior $\hat{x}_0$ is modeled as a reverse diffusion process:

$$p_\theta(\hat{x}_0|x_1) = \prod_{i=1}^{T} p_\theta(\hat{x}_{s_{i-1}}|\hat{x}_{s_i}),$$ (2)

where $\{s_i\}_{i=0}^{T}$ is an increasing sequence of diffusion steps satisfying $0 = s_0 < s_1 < \cdots < s_T = 1$, with $\hat{x}_{s_T} = x_1$ and $T$ is the number of decoding steps.

## 3.2 OVERVIEW

In this section, we introduce DIVA, a unified framework that integrates action generation into a discrete diffusion model for vision-language-action (VLA) tasks. As illustrated in Fig. 2, the architecture of DIVA consists of two innovative modules, each contributing to robust and coherent action generation. First, the Discrete Action Tokenization process (Sec. 3.3) serves as a bridge between continuous action trajectories and the discrete token space of large language models. By quantizing actions into semantically meaningful tokens, it preserves spatiotemporal structure in action and ensures native compatibility with language-guided reasoning. Second, Latent-Driven Policy Learning (Sec. 3.4) enables mutual enhancement between language representations and action decoding through a multi-objective training scheme. Finally, we introduce a Selective Group Unmasking strategy (Sec. 3.5) as an extension of discrete diffusion decoding strategies in vision-lanaguage-action tasks, which facilitates the inter-action dependencies through group-level unmasking. Together, these innovations establish DIVA as an effective framework for generating precise and temporally-coherent actions for VLA tasks.

## 3.3 DISCRETE ACTION TOKENIZATION

We formalize action representation through a Discrete Action Tokenization process, which centers on an action tokenizer built upon a Vector-Quantized Variational Auto-Encoder (VQ-VAE). This module learns an adaptive mapping from continuous motion sequences to a structured discrete vocabulary, enabling more precise and semantically meaningful encoding than current bin-based quantization approaches Kim et al. (2024).

The tokenizer extends the vocabulary codebook of a pre-trained Visual Language Model (VLM) to ensure native compatibility with large language backbones. As illustrated in Fig. 2, the encoding process begins by segmenting the input continuous action sequence along the temporal dimension. Each segment is augmented with time and action-type embeddings to preserve temporal and categorical context. The segmented actions are further split along feature dimensions and processed by a convolutional encoder $\mathcal{E}_{tok}$ to produce latent action representations. These latents are then quantized via nearest-neighbor lookup in the VLM-derived codebook, mapping them to discrete tokens. During decoding, a convolutional decoder $\mathcal{D}_{tok}$ reconstructs the original continuous action from the quantized tokens, ensuring minimal reconstruction error and high fidelity.

The action tokenizer is optimized using a VQ-VAE objective that combines the reconstruction loss, vector quantization loss, and commitment loss. The overall loss $\mathcal{L}_{token}$ is formulated as:

$$\mathcal{L}_{token} = \text{MSE}(\mathbf{S}_A, \mathcal{D}_{tok}(\mathbf{z}_q)) + \text{MSE}(\text{sg}[\mathcal{E}_{tok}(\mathbf{S}_A)], \mathbf{e}_k) + \beta \cdot \text{MSE}(\mathcal{E}_{tok}(\mathbf{S}_A), \text{sg}[\mathbf{e}_k]) \tag{3}$$

where $\mathbf{S}_A$ denotes the original continuous action, $\mathbf{z}_q$ represents the quantized latent code, $\mathbf{e}_k$ is the codebook embedding, $\text{MSE}$ denotes the mean square error, $\text{sg}[\cdot]$ denotes the stop-gradient operator introduced in VQ-VAE Van Den Oord et al. (2017), and $\beta$ controls the commitment weight.

### 3.4 LATENT-DRIVEN POLICY LEARNING

We introduce Latent-Driven Policy Learning as a joint optimization strategy to address the common disconnect between pretrained vision-language backbones and policy heads in VLA training. As illustrated in Fig. 2, this strategy employs multi-level regularization to align discrete visual-language representations with continuous actions, enabling mutual enhancement between the VLM's representative space and the policy head's action space.

The Latent-Driven Policy Learning operates by applying distinct regularization mechanisms to latent representations throughout the forward process. Given the natural language instruction $\mathbf{I}$, visual observation $\mathbf{V}$, and ground-truth action sequence $\mathbf{S}_A$, they are first encoded into discrete tokens $\mathbf{H}_I$, $\mathbf{H}_V$, and $\mathbf{H}_A$ using separate modality-specific encoders. The action tokens are randomly masked at ratio $t$ to form $\mathbf{H}_A^M$, concatenated with the visual and textual tokens, and fed into the diffusion language model to produce action hidden states $\mathbf{H}_A'$. These hidden states are then decoded by a policy head into continuous action predictions $\mathbf{O}_A$. Throughout this propagation, latent regularization is applied at multiple levels to maintain alignment and fidelity across modalities.

The regularization consists of three complementary objectives designed to jointly constrain the latent and output spaces. First, a cross-entropy loss $\mathcal{L}_{ce}$ is applied to the predicted token distribution to ensure accurate discrete reasoning:

$$\mathcal{L}_{ce} = -\mathbb{E}_t \left[ \frac{1}{t} \sum_{i=1}^{L} \mathbf{1}[\mathbf{H}_A^M = [\text{MASK}]] p_\theta(\mathbf{H}_A) \cdot \log(\mathbf{M}_c) \right], \tag{4}$$

where $\mathbf{M}_c$ denotes the confidence score matrix, and $\mathbf{1}[]$ is the indicator function.

Second, a reconstruction loss $\mathcal{L}_{recon}$ regularizes the action tokenizer by mapping hidden states back to the action codebook:

$$\mathcal{L}_{recon} = \text{MSE}(\hat{\mathbf{S}}_A, \mathbf{S}_A) + \text{MSE}(\text{sg}[\mathbf{H}_A], \mathbf{H}_A') + \text{MSE}(\mathbf{H}_A, \text{sg}[\mathbf{H}_A']), \tag{5}$$

where $\hat{\mathbf{S}}_A = \mathcal{D}_{lat}(\mathbf{H}_A)$ is the reconstructed action sequence, $\text{MSE}$ is the mean square error, and $\text{sg}[]$ denotes the stop-gradient operator.

Finally, a regression loss $\mathcal{L}_{reg}$ directly supervises the policy output:

$$\mathcal{L}_{reg} = \text{MSE}(\mathbf{S}_A, \mathbf{O}_A). \tag{6}$$

The complete training objective integrates these losses as:

$$\mathcal{L}_{total} = \mathcal{L}_{ce} + \mathcal{L}_{recon} + \mathcal{L}_{reg}. \tag{7}$$

By simultaneously optimizing discrete token prediction, latent-space reconstruction, and continuous action regression, this multi-level regularization strategy promotes a semantically consistent alignment between language-guided understanding and actionable motion generation. The approach bridges the representational gap between pretrained VLMs and policy heads, enabling more reliable action decoding.

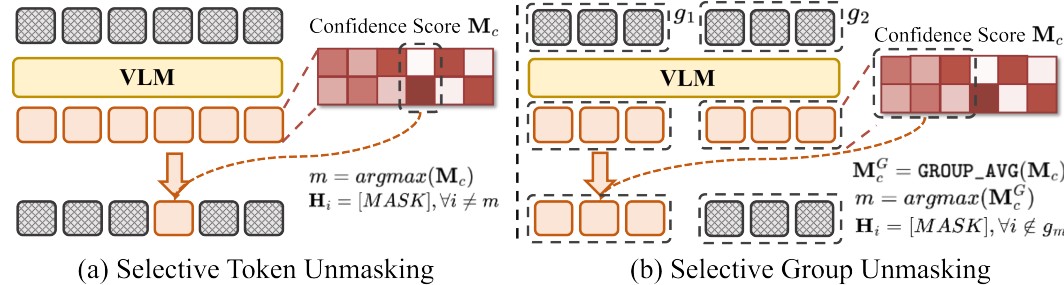

(a) Selective Token Unmasking      (b) Selective Group Unmasking

Figure 3: The comparison between (a) Selective Token Unmasking (STU) and (b) Selective Group Unmasking (SGU): STU only refines one token per step, which potentially fragments action sequences. In contrast, SGU refines the entire token groups simultaneously, preserving action dependencies to generate more coherent and physically reasonable motions.

### 3.5 SELECTIVE GROUP UNMASKING

Apart from the confidence-first principle of discrete diffusion decoding strategies Nie et al. (2025); Yang et al. (2025), we propose a Selective Group Unmasking (SGU) strategy, which advances the decoding process by shifting from a token-level to a group-level paradigm, where iterations are performed over non-overlapping groups of action tokens. This mechanism incorporates inter-action dependencies in the generation of action sequence, ensuring temporally coherent and structurally sound motions for dynamic environments.

As depicted in Fig. 3, unlike conventional Selective Token Unmasking (STU) that refines one highest-confidence token at a time, the Selective Group Unmasking (SGU) strategy operates at a grouped scale. Starting from a fully-masked sequence $\mathbf{x}$ of $L$ tokens, the diffusion large language model first produces action hidden states $\mathbf{H_x}$, which encode motion information. The corresponding token-level confidence matrix $\mathbf{M}c$ is then derived from $\mathbf{H_x}$ as the probabilities for candidate predicted tokens. The $L$ tokens are divided into $G$ non-overlapping groups $\{g_i\}_{i=1}^{G}$, each containing $K = L/G$ consecutive tokens. The token-level confidence scores are aggregated into group-level confidence scores $\mathbf{M}_c^G$ via a non-overlapping average pooling with stride $K$. Based on the score, the group with the highest aggregate confidence is then fully unmasked, while all other groups remain masked for the next refinement cycle. This process is formulated as:

$$\mathbf{M}_c^G = \texttt{GROUP\_AVG}(\mathbf{M}_c) = [u_1, u_2, ..., u_L], \quad \text{where} \quad u_i = \frac{1}{K}\sum_{i=1}^{K} max(\mathbf{M}_{c,(i-1)K+i}) \quad (8)$$

$$m = argmax(\mathbf{M}_c^G) \quad (9)$$

$$H_i = [\texttt{MASK}], \quad \forall i \notin g_m, \quad (10)$$

where $\texttt{GROUP\_AVG}$ denotes the non-overlapping average pooling operation.

By unmasking coherent groups of action tokens simultaneously, SGU balances the confidence-first principle with the intrinsic inter-action dependencies. This ensures the coherence of the generated motion, making it particularly suitable for continuous action decoding in VLA tasks.

## 4 EXPERIMENT

### 4.1 SETUP

We evaluate our approach on the LIBERO benchmark (Liu et al., 2023), which provides a diverse and standardized environment for studying embodied manipulation. The benchmark comprises four suites (LIBERO-Spatial, LIBERO-Object, LIBERO-Goal, and LIBERO-Long) with 10 tasks and 500 expert demonstrations in each suite, covering spatial reasoning, object-centric manipulation, goal-directed behavior, and extended-horizon execution. For each task, the policy observes RGB inputs from a third-person camera and a wrist-mounted camera, accompanied by a natural language

Table 1: Evaluation results on LIBERO, measured by success rate (%). Best in **bold**.

| Method | LIBERO-Sp | LIBERO-Obj | LIBERO-Goal | LIBERO-Long | Average |
|---|---|---|---|---|---|
| Diffusion Policy | 78.3 | 92.5 | 68.3 | 50.5 | 72.4 |
| Octo | 78.9 | 85.7 | 84.6 | 51.1 | 75.1 |
| DiT Policy | 84.2 | 96.3 | 85.4 | 63.8 | 82.4 |
| OpenVLA | 84.7 | 88.4 | 79.2 | 53.7 | 76.5 |
| OpenVLA-OFT | 95.2 | 94.2 | 95.2 | 93.2 | 94.5 |
| MDT | 78.5 | 87.5 | 73.5 | 64.8 | 76.1 |
| $\pi_0$ + FAST | 96.4 | 96.8 | 88.6 | 60.2 | 85.5 |
| $\pi_0$ | 96.8 | **98.8** | 95.8 | 85.2 | 94.2 |
| DIVA (Ours) | **98.0** | **98.8** | **97.6** | **95.2** | **97.4** |

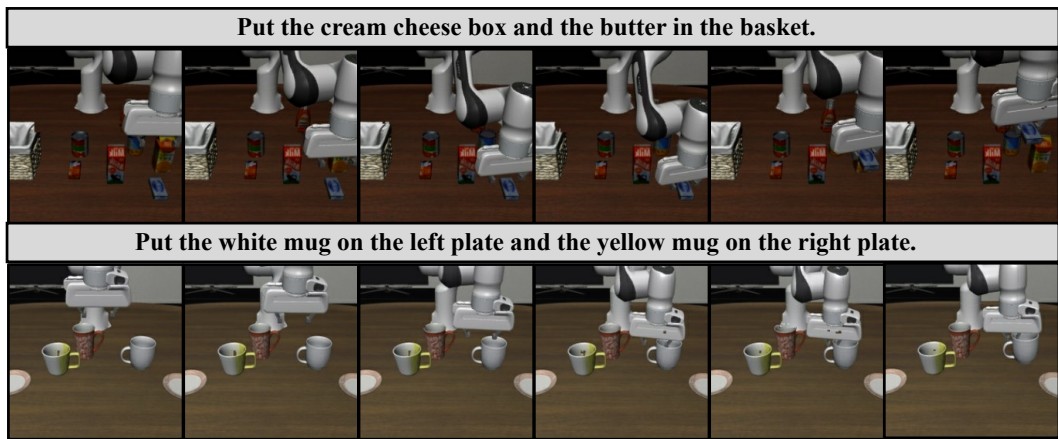

Figure 4: Demonstrations of DIVA on the LIBERO benchmark. Videos are sampled from long-horizontal tasks in the task suite of LIBERO-Long.

instruction and end-effector state. Depth images, affordance maps, and other auxiliary sensory channels are intentionally excluded to focus on learning directly from visual and language signals.

For fair comparisons, we choose a broad set of established methods spanning the two dominant paradigms of action generation. For the autoregressive baselines, we include OpenVLA (Kim et al., 2024), Octo (Team et al., 2024), OpenVLA-OFT (Kim et al., 2025), and $\pi_0$+FAST (Black et al., 2024; Pertsch et al., 2025). For the continuous diffusion and flow-matching baselines, we evaluate Diffusion Policy (Chi et al., 2023), MDT (Reuss et al., 2024), DiT Policy (Hou et al., 2025), and $\pi_0$ (Black et al., 2024). All methods are assessed under identical observation modalities and the official LIBERO evaluation metrics. Baseline results are drawn directly from the original publications or reproduced from open-source implementations to ensure consistency.

## 4.2 EVALUATION PERFORMANCE

We present qualitative performance results on the LIBERO benchmark in a simulated environment. As shown in Tab. 1, DIVA achieves superior performance with success rates of 98.0%, 98.8%, 97.6%, and 95.2% across the four task suites, yielding a top-tier average of 97.4%. Our method demonstrates a particular advantage on long-horizon tasks, outperforming the second-best method on LIBERO-Long by +2.0%. We attribute this performance gain to DIVA's structured representation space and learning strategy. The discrete action tokenization process effectively captures fine-grained motion patterns through the adaptive codebook, while the latent-driven policy learning ensures the alignment between linguistic instructions and action sequences through multi-level regularization. This combination enables temporally-coherent action modeling and precise motion execution, which is critical in complex, multi-stage scenarios. Furthermore, Fig. 4 provides quali-

Table 3: Performance of Discrete Diffusion Decoding, measured by success rate (%). Best in **bold**.

| Strategy | PD | | STU | | SGU | |
|---|---|---|---|---|---|---|
| Policy Head | CNN | DP | CNN | DP | CNN | DP |
| LIBERO | 95.2 | 95.8 | 97.0 | 96.9 | 97.2 | **97.4** |

Table 4: Real-world evaluation, measured by success rates (%). Best in **bold**.

| Method | Stack Two Cubes | Pick Up Food | Put Food in Basket | Put Food on Plate | Average |
|---|---|---|---|---|---|
| $\pi_0$ | 30 | 60 | 45 | 45 | 45 |
| DIVA (Ours) | **45** | **70** | **60** | **65** | **60** |

tative demonstrations by showcasing long-horizon tasks where DIVA capably executes multi-stage plans to achieve the desired goals, further validating the framework's ability to maintain coherent action sequences over extended time horizons.

## 4.3 ABLATION STUDY

We conduct extensive ablation studies to evaluate the contribution of each component in DIVA. As summarized in Tab. 2, we begin with OpenVLA equipped with a CNN policy head as the baseline (Model 1) and incrementally integrate our proposed modules. First, introducing the discrete action tokenizer brings a consistent improvement, raising performance by +1.4% with the CNN head (Model 2) and +1.5% with the DP head (Model 3), validating the importance of adaptive and semantically-grounded action discretization. Then, the incorporation of the Latent-Driven Policy Learning (LDPL) strategy yields additional gains of +0.6% (Model 4) and +0.7% (Model 5), demonstrating its effectiveness in aligning mul-

Table 2: Ablation study of DIVA, measured by success rate (%) on the LIBERO benchmark. Best in **bold**. [Keys: DAT: Discrete Action Tokenization; LDPL: Latent-Driven Policy Learning; CNN: Convolutional Neural Network; DP: Diffusion Policy.]

| Models | Learning Strategy | | Policy Head | | Average |
|---|---|---|---|---|---|
| | DAT | LDPL | CNN | DP | |
| 1 | | | ✔ | | 95.2 |
| 2 | ✔ | | ✔ | | 96.6 |
| 3 | ✔ | | | ✔ | 96.7 |
| 4 | ✔ | ✔ | ✔ | | 97.2 |
| 5 | ✔ | ✔ | | ✔ | **97.4** |

timodal representations and enabling more consistent action decoding through joint latent-space regularization. These results confirm that both the discrete action tokenization and the LDPL strategy are essential to DIVA's performance.

## 4.4 DISCRETE DIFFUSION DECODING

We conduct a comprehensive comparison of discrete diffusion decoding strategies to determine the optimal paradigm for action generation. Three distinct approaches are evaluated: Parallel Decoding (PD), which unmasked all tokens simultaneously; Selective Token Unmasking (STU), which unmasked individual tokens following a confidence-first principle; and our proposed Selective Group Unmasking (SGU), which unmasked coherent token groups based on aggregated confidence scores. As summarized in Tab. 3, SGU achieves consistent performance gains with both CNN and Diffuser (DP) policy heads. Specifically, it surpasses PD by +2.0% and +1.6%, and outperforms STU by +0.2% and +0.5% using CNN and DP heads, respectively. These results indicate that SGU strikes a balance between effectiveness and efficiency. It avoids the overly coarse approximation of PD while mitigating the fragmentation and error accumulation inherent in STU's sequential token-by-token unmasking. The group-wise refinement preserves local temporal dependencies within action segments, leading to more coherent and stable motion generation.

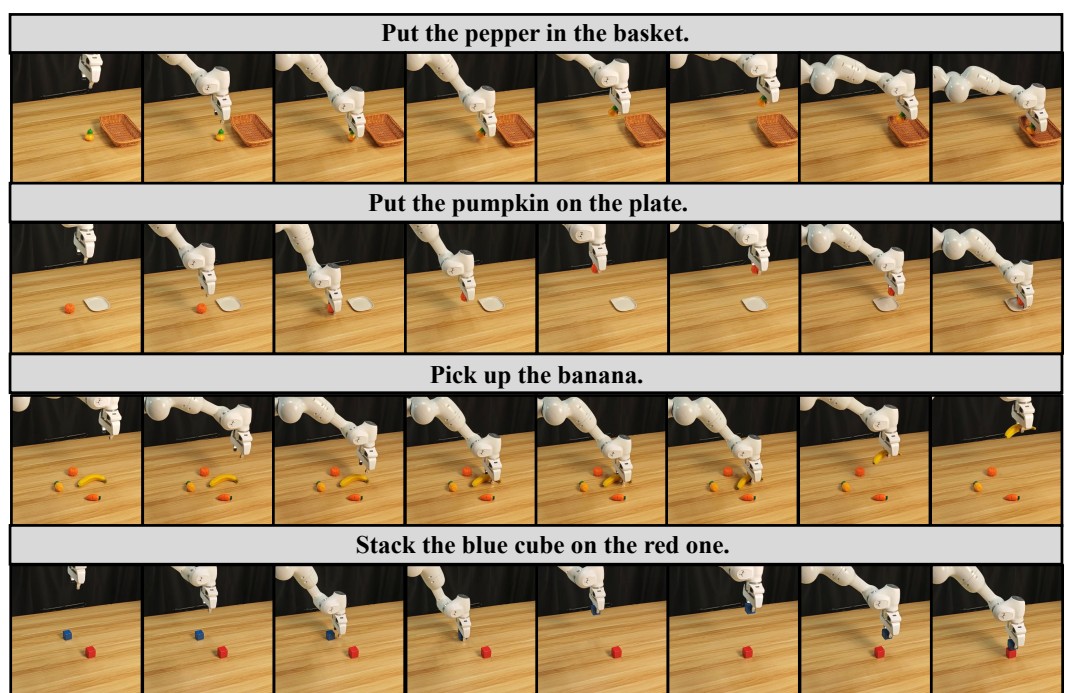

Figure 5: Demonstrations of DIVA on the real-world evaluation.

## 4.5 REAL-WORLD EVALUATION

To further validate the effectiveness of DIVA, we conduct real-world experiments using one Franka Research 3 robotic arm. The robot is equipped with an Intel RealSense D435 RGB-D camera, which provides visual observations of the workspace. The camera is mounted in a fixed position to capture both objects and the end-effector, ensuring reliable visual feedback for grasping and manipulation. The control interface is realized through the Franka Control Interface (FCI) based on the Deoxys library Zhu et al. (2022), enabling low-latency execution of continuous actions predicted by our policy. We evaluate DIVA on four representative manipulation tasks, which cover both object grasping and precise placement. The detailed descriptions of these tasks are provided in the Appendix C.1.3.

The real-world efficacy of DIVA is evaluated through qualitative demonstrations (Fig. 5) and quantitative comparisons with the baseline $\pi_0$ Black et al. (2024) (Tab. 4). The quantitative results substantiate a consistent performance gain across all tasks, raising the average success rate from $45\%$ to $60\%$. These results demonstrate that our method delivers substantial improvements in real-world manipulation, proving the effectiveness of our method under complex real-world environments.

## 5 CONCLUSION

In this paper, we present DIVA, a discrete diffusion framework for vision-language-action (VLA) tasks that reformulates action generation as a structured denoising process. Our approach introduces three key innovations: a discrete action tokenizer that adaptively maps continuous motions to semantic tokens, a latent-driven policy learning method that aligns multimodal representations with action outputs, and a selective group unmasking strategy that maintains temporal coherence during decoding. Extensive experiments show that DIVA achieves state-of-the-art performance in both simulated and real-world environments, demonstrating significant advantages in long-horizon tasks. This work establishes a new paradigm for VLA models and provides a solid foundation for future research on scalable embodied intelligence.

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

# Supplementary Material

## A  ETHICS STATEMENT

In this paper, we propose a unified vision-language-action model, DIVA, which is built upon the open-source framework OpenVLA and trained on the LIBERO, a public dataset in the simulation environments. As our research aims at advancing visual-language-action models, it does not involve the use of any personal data or human subjects. Consequently, we identify no foreseeable issues pertaining to privacy infringement, physical safety, algorithmic fairness, or other ethical conflicts. This work is conducted in full compliance with standard ethical guidelines.

## B  REPRODICIBILITY STATEMENT

Our methodology and experiments are designed to be fully reproducible. To support this goal, we will publicly release the complete source code for the DIVA model, encompassing all components for network architecture, training, and evaluation. The final trained model weights will be made available on a community platform once the paper is accepted. All datasets used in this study are publicly accessible benchmarks. Comprehensive implementation details, including hyperparameter values, training procedures, and environment setups, are described within the paper and its supplementary materials. We ensure these resources allow for the straightforward replication of our reported results.

## C  USAGE OF LLMS

In this work, large language models were utilized strictly as an auxiliary tool in two ways: firstly, for polishing the manuscript's language and enhancing its readability; and secondly, for providing technical support for the software development. The models played no role in the core research process, such as the conception of ideas, the design of the methodology, or the execution of experiments. All AI-generated content was reviewed and edited by all the authors, who take full responsibility for the presentation of the work.

### C.1  IMPLEMENTATION DETAILS

#### C.1.1  ARCHITECTURE

We develop DIVA on the foundational VLA model OpenVLA Kim et al. (2024), which consists of a language model Prismatic-7B Karamcheti et al. (2024) and two visual encoders (SigLIP Tschannen et al. (2025) and DINO v2 Oquab et al. (2023). The discrete latent action tokenizer shares the codebook with OpenVLA, with the codebook size $1024$ and the dimension $4096$. An additional `[MASK]` token is extended for discrete diffusion. We choose Convolutional Neural Network (CNN) and Diffusion Policy Chi et al. (2023) (DP) as the policy heads.

#### C.1.2  TRAINING AND INFERENCE

The training strategy of our method is a two-stage phases: 1) the cold start of discrete latent tokenization and 2) an overall finetuning of the overall architecture. In the first stage, the action tokenizer is trained with the batch size $2e-5$ and the learning rate $2e-5$ until converged. In the second stage, our method is finetuned on 8x A100 with the batch size 16 and the learning rate $2e-5$. We finetune the CNN policy head for 150K-200K iterations and the diffusion policy head for 300K-400K iterations. The sequence length of action $L$ and the masking ratio $\alpha$ are set as 8 and 0.5. During inference, the group number for SGU is set as 4.

#### C.1.3  REAL-WORLD EXPERIMENT

We conduct our real-world evaluation on a manipulation platform equipped with a single Franka Research 3 arm mounted on a fixed base. An Intel RealSense D435 RGB-D camera is placed beside the manipulator to capture the workspace, and the camera is extrinsically calibrated to the robot

base frame to enable accurate localization. The DIVA receives the raw RGB observations and the natural language instruction, and autoregressively predicts low-level robot actions in the form of end-effector deltas and gripper commands. These actions are executed step by step through the Franka Control Interface, which closes the loop between perception, reasoning, and control. Specifically, we use the Deoxys library Zhu et al. (2022) to interact with the Franka Control Interface.

In the real-world evaluation, we design four representative tabletop manipulation tasks to test the generality of our method. The tasks include: (1) Put the pepper in the basket, where the robot must grasp a small food item and place it into a container; (2) Put the pumpkin on the plate, which requires placing the target object precisely on a flat surface; (3) Pick up the banana, a grasping-only task that evaluates object detection and grasp stability under clutter; and (4) Stack the blue cube on the red one, which involves geometric reasoning and accurate placement for block stacking. These tasks cover a diverse range of skills—from basic grasping to precise placement and spatial reasoning—and their visual appearances are illustrated in Figure 5.

## D  LIMITATIONS AND FUTURE WORK

**Limitations**  : Our work presents a limitation in its reliance on the iterative refinement process of diffusion decoding, which requires multiple unmasking steps to generate high-quality actions. This refinement process inherently demands greater computational resources to achieve optimal performance, potentially hindering its application in scenarios requiring real-time decision-making.

**Future Work**  : A promising direction for future work is to develop strategies that better balance the trade-off between model effectiveness and computational efficiency. This could involve exploring accelerated sampling techniques for discrete diffusion models, designing more efficient latent representations, or investigating hybrid decoding strategies that maintain high performance while enabling faster, and potentially real-time, action prediction.

