# OpenReview forum: "DIVA: Discrete Diffusion Vision-Language-Action Models for Parallelized Action Generation"
_ICLR.cc/2026/Conference — Submitted to ICLR 2026_

### Official Review · Reviewer_Jkaj · 2025-10-28

**Soundness:** 2
**Presentation:** 2
**Contribution:** 1
**Rating:** 4
**Confidence:** 4

**Summary:**

This paper proposes DIVA, which use discrete diffusion to generate actions. It contains three designs, including a discrete action tokenization, a policy learning, and a selective group unmasking strategy.

**Strengths:**

- The writing is easy to follow.

- The method contains a lot of careful designs.

- The performance on LIBERO & real world robot surpass the baselines.

**Weaknesses:**

- The main idea of the method is using the discrete diffusion instead of continuous diffusion for VLA models. The novelty is limited.

- The policy contains a lot of designs, including the unmasking trick, regularization trick, etc. These designs are separate from the main idea and can also be applied to other method. Together with the limited performance gain (+3% success rate), makes the source of gain unclear.

- See questions below.

**Questions:**

(1)	In L071-L073, the authors claim that: continuous diffusion models “often operate as decoupled decoders that are misaligned with the pretrained representations of the underlying vision-language backbone, leading to inefficiencies and training instability.” Is there any evidence to support this claim? How to evaluate the efficiency and training stability with of the continuous models vs other models?

(2)	In L222, the action tokenizer is quantized via nearest-neighbor lookup in the VLM-derived codebook. Just for verification, will the VLM use the updated codebook also for the language part or just for the action part? Is the VQ size the same for action and language? If so, I think the VQ size for language might be too big for action. If not, why using a shared codebook as initialization? Ablation is needed to prove this design.

(3)	For the policy learning part, what is the difference between reconstructed action $\hat{S}_A$ and the decoded action $O_A$? Both of them seem to be the output from the hidden states. The final loss seems to contain both losses ($MSE(\hat{S}_A, S_A)$ and $MSE(S_A, O_A)$).

(4)	For the OpenVLA-OFT baseline, from the official paper, the LIBERO score is 97.6 (Sp) / 98.4 (Obj) / 97.9 (Goal) / 94.5 (Long) / 97.1 (Avg).  While in the paper, the OpenVLA-OFT has an averaged success rate of 94.5. How did that score calculate? Compared with the official score, the performance gain (+0.3% success rate) seems to limited.

---

> ### Author Response · Authors · 2025-11-29
>
> We sincerely thank Reviewer Jkaj for the positive assessment of our work's design and presentations. Below, we provide extensive responses, justifications and experiments to the concerns.
>
> > Q1: The necessity of each design
>
> Our core contribution is the unified discrete diffusion framework that effectively bridges VLM representations with policy learning. The necessity and distinct contribution of each design are empirically demystified through our comprehensive ablation studies.
> The results in Tables 2 and 3 from our paper provide decompositions of the overall performance gain. As shown in Table 2, the introduction of Discrete Action Tokenization provides a foundational improvement of +1.4%, establishing a robust action representation. Building on this, our Latent-Driven Policy Learning mechanism contributes a further +0.6% gain by ensuring the policy is driven by regularized VLM latents.
>
> Furthermore, Table 3 in General Respose isolates the impact of the decoding strategy, demonstrating that our Grouped Unmasking is significant improvement, which yields gains of +2.0% and +1.6% with different policy heads (CNN and DP). This shows its critical role in fine-grained action prediction.
>
> > Q2: Analysis of Training Stability and Efficiency
>
> We provide the training standard deviation (Q1, General Response) and inference speed (Q2, General Response) to justify the training stability and efficiency of our method.
>
> > Q3: Analysis of the VQVAE design
>
> Our method employ the VLM to process both language and action part within a unified representational space derived from an updated VQ codebook. We implement this integration via a projection step. The compact and structural rich VQ hidden states (dimension 16) are mapped to the VLM's high-dimensional embedding space (dimension 4096). This approach effectively initializes the VLM's action representation space with the structural prior of the pretrained VQ-VAE. Consequently, the VLM operates on action embeddings that are suitable for fine-grained prediction, which contributes to the improved performance shown in Q3, General Response.
>
> > Q4: The Role of $\mathbf{S}_A$ and $\mathbf{O}_A$ in Policy Learning
>
> The role of $\mathbf{S}_A$ and $\mathbf{O}_A$ are different supervisions implemented in our policy learning framework.
>
> The reconstructed action ($\mathbf{S}_A$) is the quantized output of the pretrained VQ-VAE. The loss $\mathcal{L}_{recons}$ between the original and reconstructed actions ensures that the VLM's continuous hidden states remain aligned with the discrete space of the pretrained VQ-VAE codebook. This process preserves crucial structural information from the multimodal VLM space, providing a coherent foundation for the discrete diffusion model to perform token-level prediction.
>
> In contrast, the decoded action ($\mathbf{O}_A$ ) is the output of a separate action decoding head. This component serves as the final policy output for fine-grained actions, which operates on the well-regularized VLM's hidden states to generate precise action sequences. This design allows for a dedicated decoding phase that translates the model's internal representations into accurate manipulation.
>
> In conclusion, the two components serve as complementary roles. The reconstruction loss ensures the model learns structural embeddings from the pretrained quantized codebook, while the regression loss maps these representations to more precise actions. The effectiveness of this design is validated by the performance improvement by 0.5% reported in Table 2 from our paper.
>
> > Q5: Performance Comparison between OpenVLA-OFT and DIVA
>
> In Table 1 from our paper, we report the reproduced result of OpenVLA-OFT on LIBERO based on the official repository. We acknowledge that the saturated benchmark makes the improvement minor. To further validate the effectiveness of our method, we provide additional experiments on the RoboTwin benchmark in Q4, General Response.

---

### Official Review · Reviewer_oNqg · 2025-10-28

**Soundness:** 2
**Presentation:** 3
**Contribution:** 3
**Rating:** 4
**Confidence:** 4

**Summary:**

The paper introduces DIVA, a discrete diffusion-based vision-language-action (VLA) framework that addresses the misalignment between pretrained vision-language representations and downstream action learning in prior VLA methods. DIVA introduces two strategies, LDPL and SGU, and demonstrates their effectiveness through ablation studies. In both real-world and simulation experiments, DIVA outperforms the strong VLA baseline π₀, demonstrating the advantages of the discrete diffusion model.

**Strengths:**

- DIVA introduce discrete diffusion large language models into the VLA domain, leading to significant performance improvements.
- The ablation studies are comprehensive, and the proposed tricks provide noticeable performance improvements.
- Clear presentation of methodology

**Weaknesses:**

- **Relatively weak experiment**
 The paper evaluates its method only in the LIBERO simulation environment, while the real-world experiments involve relatively simple tasks and scenarios. Moreover, the absence of demonstration videos raises concerns about the model’s real-world performance.

- **Limited Empirical Evidence Supporting the Conclusions**
 Many VLA methods already achieve high performance in the LIBERO simulation environment due to its relatively simple tasks and low generalization requirements, resulting in only marginal improvements for DIVA. Moreover, in the ablation results, removing either the LDPL or SGU tricks yields performance comparable to the baseline method OpenVLA-OFT, making it difficult to substantiate the claim that DIVA effectively addresses the issue of misalignment with the pretrained representations of the underlying vision-language backbone.

- **Figure error**
 According to the description in Equation (5), $L_{\text{recon}}$ should be computed using $H_A$; however, in Fig.2 the loss is shown as being computed at $H_{A}'$.  This appears to be a minor inconsistency/error.
​

**Questions:**

1. In real-world experiments, for tasks where DIVA achieves a higher success rate than π₀,  what are the corresponding failure cases of π₀ and successful cases of DIVA?  Please provide detailed video demonstrations for comparison.
2. How do the real-world experiments demonstrate the model’s ability to handle long-horizon tasks?
3. The experiments on LIBERO seem to show that, without the two proposed tricks,  the discrete diffusion-based VLA framework performs comparably to the baseline methods  (OpenVLA, π₀, etc.).  How can the authors substantiate the claim that this framework effectively alleviates the disconnect between the pretrained backbone and the policy head? It is recommended to include comparisons on more challenging benchmarks (such as Simpler[1], Robotwin[2], or Calvin[3]) to better demonstrate the model’s effectiveness and generalization ability

[1] Li, Xuanlin, et al. "Evaluating real-world robot manipulation policies in simulation." arXiv preprint arXiv:2405.05941 (2024).

[2] Mu, Yao, et al. "Robotwin: Dual-arm robot benchmark with generative digital twins (early version)." European Conference on Computer Vision. Cham: Springer Nature Switzerland, 2024.

[3] Mees, Oier, et al. "Calvin: A benchmark for language-conditioned policy learning for long-horizon robot manipulation tasks." IEEE Robotics and Automation Letters 7.3 (2022): 7327-7334.

**Details Of Ethics Concerns:**

Please see above weaknesses and questions

---

> ### Author Response · Authors · 2025-11-29
>
> We sincerely thank Reviewer oNqg for the positive assessment of our work's motivation and presentations. Below, we provide extensive responses, justifications and experiments to the concerns.
>
> > Q1: Extensive Real-World Evaluation
>
> We provide the evaluation in Q5, General Response.
>
> > Q2: Extensive Simulation Evaluation
>
> We provide the evaluation in Q4, General Response.
>
> > Q3: Justification of Alignment between the Pretrained Backbone and the Policy Head
>
> We provide the justification in Q6, General Response.
>
> > Q4: Figure Error
>
> We thank the reviewer for this thoughtful observation. The representation in Fig. 2 is correct and reflects a key design choice. By computing $\mathcal{L}_{recons}$ between the embedding from the VQ codebook ($\mathbf{H}_A$) and the hidden states $\mathbf{H}_A^\prime$, we regularize the quantized representations with the continuous VLM outputs. This alignment is crucial as it preserves the structural information from the pretrained VQ-VAE codebook within the VLM's latent space, which we demonstrate is essential for generating precise action sequences in Table 2.

---

### Official Review · Reviewer_hnUQ · 2025-11-01

**Soundness:** 3
**Presentation:** 3
**Contribution:** 2
**Rating:** 4
**Confidence:** 3

**Summary:**

The paper introduces DIVA, a Discrete Diffusion Vision-Language-Action (VLA) model designed to address limitations of existing autoregressive and continuous diffusion VLA approaches.
DIVA reformulates action generation as a discrete denoising diffusion process operating over quantized latent action tokens.

**Strengths:**

1. The paper’s discrete diffusion formulation for action generation is novel.
2. Selective Group Unmasking is a good design and is verified in experiments.

**Weaknesses:**

1. The real-world experiments are relatively simple and do not adequately demonstrate the policy’s performance in real-world settings.

2. The authors claim that autoregressive-style VLAs suffer from error accumulation and temporal rigidity in action generation, but they do not provide specific experiments to directly validate these claims.

**Questions:**

1. The authors claim that autoregressive-style VLAs suffer from error accumulation and temporal rigidity in action generation. Are there any specific experiments that directly verify these claims?

2. Why do the authors use an additional policy head to decode the actions instead of directly decoding the tokens generated by the diffusion VLM into raw actions by VQVAE encoder? If an additional policy head is used, how can the generalization ability inherited from the VLM preserved?

---

> ### Author Response · Authors · 2025-11-29
>
> We are grateful to Reviewer hnUQ for the positive assessment of our work's motivation and design. Below, we provide extensive responses, justifications and experiments to the concerns.
>
> > Q1: Extensive Real-world Demonstrations
>
> We provide extensive demonstrations and analysis in Q5, General Response.
>
> > Q2: Comparison between Autoregressive and Discrete Diffusion Decoding on Temporal Rigidity
>
> We report extensive ablation studies and analysis in Q3, General Response.
>
> > Q3: Why Use an Additional Policy Head Instead of Decoding VQVAE Tokens?
>
> Table 3 in General Response provides the performance of multiple action decoding strategies on LIBERO. As is shown in the first row, a VLM with the VQ-VAE decoder performs significantly worse, achieving only an average success rate of 84.7%. We deem that VQ-VAE decoder is trained solely for passive trajectory reconstruction rather than task-optimal control, and lacks fine-grained adjustments for precise manipulation. In contrast, introducing a lightweight policy head (e.g., a CNN or diffusion policy head) improves performance to 95.1% and 97.4%, respectively. These results highlight that the policy head does not replace the VLM’s latent representations, but effectively refines them to produce accurate actions.
>
> > Q4: How can the generalization ability inherited from the VLM preserved?
>
> We provide the justification in Q6, General Response.

---

### Official Review · Reviewer_D3gh · 2025-11-01

**Soundness:** 3
**Presentation:** 3
**Contribution:** 3
**Rating:** 8
**Confidence:** 4

**Summary:**

The paper introduces **DIVA** (Discrete Diffusion Vision-Language-Action), a framework that reformulates **action generation** as an **iterative denoising process** over **discrete latent representations**. The proposed pipeline integrates three key components: **learnable discrete action tokenization**, **latent-driven policy learning**, and **selective group unmasking**. Extensive experiments on **LIBERO** and **real-robot** benchmarks demonstrate that **DIVA** achieves superior performance compared to existing **imitation learning** and **Vision-Language-Action (VLA)** models.

**Strengths:**

1. The paper tackles a **highly important and timely research question**, particularly as diffusion models are becoming increasingly influential in the fields of **imitation learning**, **reinforcement learning**, and **Vision-Language-Action (VLA)** modeling.

2. The paper presents **extensive experiments** in both **simulation** and **real-robot** settings, providing strong empirical validation of the proposed method’s effectiveness.

3. The paper is **well-written**, **clearly structured**, and **easy to follow**, making the technical content accessible and well-motivated.

**Weaknesses:**

1. The authors should report the **number of random seeds** and the corresponding **standard deviations** for the main results presented in **Table 1**.

2. In **Table 2** of the ablation study section, **Model 1 (OpenVLA)** achieves a performance of **95.2%**, whereas in the main results of **Table 1**, **OpenVLA** reports an average performance of only **76.5%**. The authors should clarify the reason for this **data discrepancy**.

3. What is the **inference time** of **DIVA** for generating a sequence of actions? Does the use of **discrete diffusion models** increase inference latency, potentially limiting the method’s **practical applicability in real-world settings**?

**Questions:**

1. **(Related to Weakness 2)** What explains the **performance discrepancy** of **OpenVLA** between **Table 1** and **Table 2**?

2. **(Related to Weakness 3)** What is the **inference time** of **DIVA**? Does the use of **discrete diffusion models** introduce additional **inference latency**?

---

> ### Author Response · Authors · 2025-11-28
>
> We are grateful to Reviewer D3gh for the positive assessment of our work's motivation, presentation, and contributions. Below, we provide extensive responses and experiments to the concerns.
>
> > Q1: Performance Reproducibility
>
> We report the performance in Q1, General Response.
>
> > Q2: Performance Discrepancy between OpenVLA and Our Method
>
> The performance gap between (1) OpenVLA in Table 1 and (2) Model 1 in Table 2 stems from the fact that they are two different methods.
> - (1) OpenVLA refers to the autoregressive vision-language-action model introduced in [1].
> - (2) Model 1 integrates several enhancements on the backbone, including action chunking, and discrete diffusion.
>
> Although both models share the similar model architecture, their decoding mechanisms and implementation differ substantially, leading to different performance.
>
> > Q3: Analysis of Efficiency
>
> We report the performance in Q2, General Response.
>
> **References**
>
> [1]: Kim, Moo Jin, et al. "Openvla: An open-source vision-language-action model." arXiv preprint arXiv:2406.09246 (2024).

---

### Author Response · Authors · 2025-11-29
**General Response**

We are deeply grateful to the reviewers for their rigorous review and the insightful suggestions that have been provided. The positive evaluations are much appreciated, and we provide all the following justifications to refine our work and address common concerns.

> Q1: Performance Reproducibility and Stability

**Table 1: Performance evaluated on LIBERO**

|Method|LIBERO-Sp|LIBERO-Obj|LIBERO-Goal|LIBERO-10|Average|
| -- | -- | -- | -- | -- | -- |
|Diffusion Policy| 77.92 ± 0.28 | 92.12 ± 0.2 | 67.88 ± 0.25 | 49.52 ± 0.42 | 71.86 |
|Octo| 78.76 ± 0.22 | 85.52 ± 0.20 | 84.16 ± 0.39 | 51.0 ± 0.31 | 74.84 |
|DiT Policy| 84.08 ± 0.22 | 96.0 ± 0.3 | 84.8 ± 0.27 | 63.1 ± 0.39 | 82.0 |
|OpenVLA| 84.32 ± 0.35 | 88.08 ± 0.26 | 78.32 ± 0.52 | 52.6 ± 0.43 | 75.83 |
|OpenVLA-OFT| 96.08 ± 0.28 | 96.96 ± 0.35 | 96.62 ± 0.20 | 92.92 ± 0.35 | 95.65 |
|MDT| 77.6 ± 0.3 | 86.2 ± 0.48 | 72.96 ± 0.42 | 64.12 ± 0.3 | 75.22 |
|pi_0 + FAST| 96.08 ± 0.35 | 96.52 ± 0.20 | 88.08 ± 0.27 | 59.80 ± 0.48 | 85.21 |
|pi_0| 96.12 ± 0.28 | 98.52 ± 0.16 | 95.44 ± 0.3 | 84.92 ± 0.37 | 93.75 |
|DIVA| 97.6 ± 0.25 | 98.6 ± 0.20 | 97.12 ± 0.3 | 94.8 ± 0.33 | 97.03 |

Table 1 provides detailed performance metrics with the standard deviation over five independent runs. We use random seeds 1, 10, 100, 1000, 10000 for evaluation and report the averaged results for each method. Our method achieves the best performance with an average success rate of 97.03%, demonstrating its training stability and effectiveness across runs.

> Q2: Inference Speed

**Table 2: Inference Efficiency Comparison**
|Method| Speed (Hz)|
| -- | -- |
|OpenVLA-OFT + AR| 5.84 |
|OpenVLA-OFT + AC & PD | 29.52 |
|DIVA (ours) | 21.04 |

Table 2 reports the inference speed of our method and OpenVLA-OFT on an H200 GPU. Our method runs at 21.04 Hz, significantly faster than the 5.84 Hz autoregressive (AR) version. This gain primarily comes from the simultaneous decoding enabled by discrete diffusion, which substantially accelerates forward propagation.

Although our method achieves 0.71× the speed of OpenVLA-OFT with parallel decoding, the diffusion-based multi-round unmasking refinement introduces a 2.9% absolute improvement in LIBERO success rate. We consider this tradeoff favorable, as the current inference speed remains within the range for real-world robotic deployment.

Moreover, recent advances in diffusion acceleration [1, 2] provide additional mechanisms for further reducing computational overhead. We believe discrete diffusion-based policies will naturally benefit from these developments, leading to more efficient deployment in the future.

> Q3: Ablation Studies of Multiple Action Decoding Strategies

**Table 3: Comparison between Autoregressive and Discrete Diffusion Decoding**

|Method|LIBERO-Sp|LIBERO-Obj|LIBERO-Goal|LIBERO-10|Average|
| -- | -- | -- | -- | -- | -- |
|OpenVLA w/AR | 88.6 | 92.4 | 85.2 | 72.6 | 84.7 |
|OpenVLA w/Block-wise Unmasking| 94.4 | 97.8 | 95.2 | 93.0 | 95.1 |
|OpenVLA w/Selective Group Unmasking| 98.0 | 98.8 | 97.6 | 95.2 | 97.4 |

To directly examine the impact of decoding strategy on temporal rigidity, we compare three inference patterns on the OpenVLA backbone.

 (1) Autoregressive (AR) decoding: tokens are generated strictly one-by-one;

 (2) Block-wise unmasking: tokens within each block are decoded in parallel, block-wise tokens are decoded sequentially;

 (3) Selective group unmasking: groups of tokens are decoded based on confidence scores (details in Sec. 3.5).

 For both (2) and (3), we maintain a fixed action chunk length of $L = 16$. We evaluate all these patterns on the LIBERO benchmark.
As shown in Table 3, the result demonstrates that autoregressive decoding exhibits lower performance across all LIBERO subsets, with an average of 84.7%, significantly below the block-wise unmasking (95.1%) and selective group unmasking (97.4%) ones. This performance gap directly reflects two limitations of the AR decoding paradigm:

1. Error Accumulation: AR decoding is inherently sequential. Any local prediction error propagates to all subsequent actions. In manipulation tasks requiring long-horizon precision (e.g., LIBERO-10), this will result in more degraded performance (72.6% vs. 93.0% / 95.2%) than discrete diffusion backbones.

2. Temporal Rigidity: AR decoding enforces a fixed temporal order and prevents the model from revising earlier action tokens once generated. In contrast, discrete diffusion unmasking allows joint refinement of previous tokens, enabling the model to modify earlier suboptimal predictions.

These experiments provide evidence for our claim that autoregressive VLAs suffer from temporal rigidity and error accumulation, while discrete diffusion models mitigates these issues and leads to better performance.

**References**

[1] Wu, Chengyue, et al. "Fast-dllm v2: Efficient block-diffusion llm."

[2] Zheng, Haoyang, et al. "Ultra-fast language generation via discrete diffusion divergence instruct."

---

> ### Author Response · Authors · 2025-11-29
> **General Response**
>
> > Q4: Extensive Simulation Evaluation
>
> **Table 4: Performance evaluated on RoboTwin**
>
> |Method|Beat Block Hammer|Pick Diverser Bottles|Put Object Cabinet|Move Pillbottle Pad|Average|
> | -- | -- | -- | -- | -- | -- |
> |OpenVLA-OFT | 37% | 28% | 38% | 16% | 29.75% |
> |DIVA (Ours)| 48% | 42% | 46% | 28% | 41% |
>
> To further support our method, we conduct extensive simluation evaluation on the RoboTwin[3] benchmark. Table 4 reports results across four representative tasks. As is shown, OpenVLA-OFT reaches an average success rate of 29.75%, while our method (DIVA) improves the overall performance to 41.0%. These improvements appear in skills that require precise contact interaction in Beat Block Hammer, multi-object manipulation in Pick Diverse Bottles, constrained placement in Put Object Cabinet, and subtle motion refinement in Move Pillbottle Pad. The consistent gains across all tasks indicate that the discrete diffusion architecture with our decoding strategies allows the policy to integrate the pretrained multimodal backbone more effectively and to predict actions based on the underlying latent representations. The evaluation on RoboTwin demonstrates that the proposed framework generalizes to a broader set of manipulation scenarios and provides strong evidence for its effectiveness.
>
> > Q5: Extensive Real-World Evaluation
>
> We provide extensive video demonstrations including both simple operation tasks and multi-stage tasks. The demonstrations are accessible via this anonymous link: https://anonymous.4open.science/r/DIVA-demos/
>
> **Table 5: Extensive real-world performance**
>
> |Method|Stack Two Cubes | Pick Up Food | Put Food in Basket | Put Food on Plate | Pick and Place Two Food | Stack Three Cubes | Average |
> | -- | -- | -- | -- | -- | -- | -- | -- |
> |$\pi_0$ | 30% | 60% | 45% | 45% | 15% | 10% | 34.2% |
> |DIVA(Ours)| 45% | 70% | 60% | 65% | 25% | 20% | 47.5% |
>
> Table 5 demonstrates that DIVA delivers substantial and consistent improvements over the baseline $\pi_0$, raising the average success rate from 34.2% to 47.5%. The performance reveals two key strengths of our method.
>
> First, on the simpler, single-stage tasks (Stack Two Cubes, Pick Up Food, Put Food in Basket, Put Food on Plate), DIVA achieves a significantly higher success rate from 10% to 20%. This indicates that our method's use of discrete tokenization and latent-driven regularization leads to more precise and reliable actions.
>
> Second, DIVA shows advantages on the long-horizon, multi-stage tasks of "Pick and Place Two Food" and "Stack Three Cubes," where its success rates are 10% better than $\pi_0$. This performance gap underscores DIVA's superior ability to maintain stable and correct action sequences over extended time horizons. We attribute this to our model's structurally coherent latent space, which mitigates error accumulation. While $\pi_0$ often fails in an early stage of the task due to imprecise manipulation, DIVA generates more stable trajectories, enabling it to reliably progress through the entire sequence of sub-tasks. This robust long-horizon performance is a critical validation of our approach's efficacy in complex, real-world environments.
>
> > Q6: Justification of Alignment between the Pretrained Backbone and the Policy Head
>
> The introduction of policy heads does not diminish the generalization ability of the VLM. The VLM remains responsible for generating the high-level discrete action representations, which encode the multimodal priors learned from large-scale pretraining. The policy head serves as a lightweight decoder that maps these latent tokens into precise control commands. Owing to its low capacity and its dependence on the VLM’s outputs, the policy head cannot override or replace the VLM’s inherent prior knowledge. In addition, as demonstrated in the substantial performance gains in Table 2 from our paper, the policy head enables the model to better exploit the VLM’s generalizable action structure while providing the fine-grained refinement for accurate robotic manipulation.
>
> Furthermore, the reconstruction loss $\mathcal{L}_{recon}$ introduced in Sec. 3.4 regularizes the latent representation, ensuring consistency between the pretrained action codebook and the multimodal spaces of the VLM. This additional constraint further supports the preservation and integration of the VLM’s generalization capability within the overall policy.
>
> **References**
>
> [3] Mu, Yao, et al. "Robotwin: Dual-arm robot benchmark with generative digital twins." Proceedings of the Computer Vision and Pattern Recognition Conference. 2025.

---

### Meta-Review · Area_Chair_1PME · 2026-01-10

**Summary:**

This paper proposes a discrete diffusion vision-language-action model, which is one of the hottest topics. As pointed out by all reviewers, the paper is well-written and easy to follow.

However, most reviewers raised several serious concerns and submitted negative ratings. The concerns include: (1) lacking experiments on different simulation environments, (2) limited performance gain with plenty of components, (3) too simple real-world experiments, and (4) unconvincing motivation.

Although the authors made great efforts during the rebuttal phase, the three reviewers didn't increase the scores. I'm inclined to recommend it for rejection.

**Reviewer Concerns:**

The authors supplemented a lot of experiments.

**Reviewer Scores:**

No, the three reviewers may insist their initial rating.

---

### Decision · Program_Chairs · 2026-01-26

Reject